# French validation of the Weight Efficacy Life-Style questionnaire (WEL): Links with mood, self-esteem and stress among the general population and a clinical sample of individuals with overweight and obesity

**Natalija Plasonja**[1]*, **Anna Brytek-Matera**[2], **Greg Décamps**[1]

**1** LabPsy, University of Bordeaux, Bordeaux, France, **2** Institute of Psychology, University of Wroclaw, Wroclaw, Poland

☯ These authors contributed equally to this work.

* natalija.plasonja@u-bordeaux.fr

**Data Availability Statement:** The data analyzed in this study cannot be shared publicly due to privacy

## Abstract

Eating self-efficacy refers to a person's belief in their ability to regulate eating. Although the Weight Efficacy Life-Style questionnaire (WEL) is one of the most widely used eating self-efficacy tools, its French validation is lacking. The objective of this research was to validate a French version of the WEL in a general and a clinical sample, and to explore the links between eating self-efficacy and psychosocial variables. In study 1, the general population sample included 432 adults (93% of women, mean age = 43.18 ± 11.93 years). In study 2, the clinical sample included 2010 adults with overweight and obesity (87% of women, mean age = 44.44 ± 11.25 years). Exploratory and confirmatory factor analyses were performed. Two distinct versions of the WEL were retained: a 12-item questionnaire intended for use in the general population, named WEL-Fr-G, and an 11-item questionnaire for clinical samples, named WEL-Fr-C. The two French versions of the WEL presented strong reliability and sensibility. In addition, study 2 provided support for the measurement invariance of the WEL-Fr-C across sex and Body Mass Index. The two versions are therefore psychometrically sound instruments for assessing eating self-efficacy in the general population (WEL-Fr-G) and clinical samples (WEL-Fr-C).

## 1. Introduction

Obesity is defined by the World Health Organization (WHO) as "the abnormal or excessive fat accumulation that presents a risk to health" [1]. Overweight and obesity constitute a major risk factor for the onset of severe health complications such as cardiovascular and respiratory diseases, diabetes and some types of cancers [1, 2], hence making it one of the major challenges for public health on a global scale. In France, nearly 50% of the adult population was overweight or suffered from obesity in 2015 [3]. The main cause of overweight and obesity relies in

concerns. The data access requests should be addressed to R&D Innovation Director at PiLeJe, Michel Dubourdeaux (m.dubourdeaux@pileje.com).

**Funding:** Our study was co-funded by PiLeJe Laboratoire, within the framework of the CIFRE convention, grant number 2019/0009, and the „Excellence Initiative – Research University" program for years 2020-2026 of the University of Wroclaw. The funders had no role in study design, data collection and analysis, decision to publish, or preparation of the manuscript.

**Competing interests:** The authors have declared that no competing interests exist.

an energy imbalance between caloric intake and expenditure [4–6]. Reasons behind this imbalance are various and include individual's lifestyle [6, 7], physiological causes (such as the microbiome [8] or sleep duration [9, 10]) but also individual's socio-economic situation [11], obesogenic environments such as the proximity to fast-food restaurants or a gym [12–15], genetics [16] and even air pollution [17]. Numerous psychological factors of obesity were identified as well [5]. Yet, treatment that is mostly offered to patients suffering from obesity consists of life style changes embodying the adoption of healthy eating habits and increased physical activity [18–20]. Long-term follow-up evaluations revealed that behavioural interventions, promotion of healthier eating habits and increased physical activity are not effective in treating obesity over the long term. Researchers therefore began to explore psychosocial predictors of weight regain and barriers to weight maintenance. Results from previous studies identified various behavioural correlates, such as dietary intake and physical activity as well as diverse psychological correlates, such as psychological distress, low well-being, binge-eating disorder, body dissatisfaction, low self-esteem or low quality of life [21–25]. More precisely, some authors have identified self-efficacy as one of the variables that predicts long-term behaviour change [26].

Self-efficacy is the key component of Bandura's social cognitive theory defined as individual's confidence in their ability to perform and succeed in certain situations [27, 28]. Self-efficacy is domain-specific and is, therefore, found to be a predictor of behaviours such as academic performance [29], shift to electric vehicles in Chinese adults [30] and also health behaviours like tobacco dependence [31], substance use disorders [32], diabetes-related care [33, 34], asthma self-management [35] and exercise [36]. Self-efficacy plays an important role in many theories of health behaviour change, including the Trans theoretical Model [37] and the Health Belief Model [38], because it motivates behaviour and predicts its initiation. Weight related self-efficacy is considered an important psychological factor of weight management as it predicts weight loss, weight maintenance and dietary intake. More specifically, eating self-efficacy (ESE), defined by Clark *et al.* [39] as the ability of individuals to resist their desire to eat in different situations, was found to predict dietary behaviour [40, 41]. Indeed, ESE was positively related to increased fruit and vegetable consumption [42–45] and negatively related to saturated fat and fast food consumption [44, 46]. ESE was also identified as a mediator between social support and dietary intake in low-income women [43, 46, 47] as well as a moderator between intention and eating behaviour [41]. Not surprisingly, ESE was also linked to weight loss [48]. However, the reported results are conflicting. While numerous authors have identified ESE as a predictor of weight loss [18, 45, 49–53], some authors have found opposite results: ESE at baseline was not a predictor of weight loss [54, 55]. Some studies have shown that higher ESE levels before weight loss management predicted lower weight loss, which can be interpreted as overconfidence on the part of the patient, leading to disruption of the weight loss attempt [45, 56]. However, changes in ESE, due to the weight loss treatment in which the participants were enrolled, may be a more significant predictor of weight loss than ESE at baseline [45, 54–55]. Based on the results of previous studies, ESE can also play an important role in weight maintenance. Foreyt *et al.* [22] found that weight maintainers had better ESE related to negative affect than individuals who regained weight. Similarly, Pedersen *et al.* [57] found that long-term maintainers reported a wider range of self-regulatory strategies than short-term maintainers, which also involved self-efficacy related to meal planning and shopping. It is interesting to note, however, that Clark and King [58] found no change in ESE levels in case of weight cycling. Beyond individual characteristics, some studies have identified that parental self-efficacy can influence children's food choices and their adoption of weight-protective behaviours [33, 59–61].

Positive correlations have been observed between ESE and well-being, affective balance [62], self-esteem [63] and obesity-specific quality of life after sleeve gastrectomy [50]. Conversely, some authors have reported negative correlations between ESE and symptoms of eating disorders [64–66], stress [55, 67] and depression [68, 69]. Annesi [70] even identified ESE as a mediator between depression and emotional eating.

In contrast, the links between ESE and sociodemographic variables produced inconsistent results. While some studies reported higher ESE levels in women, [18], others showed opposite results suggesting that men have higher ESE levels [22, 48, 65, 71]. However, according to Presnell, Pells, Stout and Musante [52], these gender differences disappeared after treatment. Some studies have not identified significant differences in ESE levels according to participants' gender [58, 68]. Furthermore, various studies have reported negative correlations between ESE and BMI [18, 40, 47, 51, 63, 65, 72, 73]. Only one study found that patients with increased BMI had higher ESE scores [68] and a few reported insignificant correlations between BMI and ESE [48, 52, 64]. According to Ames *et al.* [68], younger participants had higher ESE levels but no significant correlation between these two variables was found in other studies [47, 64]. Nevertheless, results involving sociodemographic variables should be considered with caution since, in a vast majority of studies, participants were predominantly women (61%– 91% female sample, sometimes even only female), overweight and/or suffering from obesity and, mostly enrolled in a weight-loss treatment. Furthermore, the majority of studies were conducted on American adults, while only few studies included European samples (Spain [62], Norway [74], Germany [41], Denmark [57] and Italy [65]). Moreover, caution should be exercised since the data in these studies were collected using different ESE questionnaires.

The first eating self-efficacy Scale was created in 1984 by Jeffery *et al.* [75]. The scale consists of 30 items: 15 items evaluating feeling self-efficacy (the person's ability to resist eating when feeling certain emotions) and 15 items regarding situational self-efficacy (resisting the urge to eat when engaging in certain activities). The development of eating-self efficacy scales has been widely expanded with more than dozens of different ESE scales available, some of them being addressed to specific populations such as low-income women [55, 76], students [77], children and their parents [33] or athletes [66]. Some scales have been developed recently to assess ESE in relation to specific diets such as Mediterranean Diet [62] or gluten-free diet for people suffering from celiac disease [71]. The Weight Efficacy Life-Style questionnaire (WEL) developed by Clark, Abrams, Niaura, Eaton and Rossi [39] is one of the most widely used. It has been used in many studies [18, 42, 51, 53, 54, 56, 68–70, 78–80] and populations such as adults suffering from cancer [51], type 2 diabetes [81] or adults with obesity and enrolled in different weight loss programs [39, 42, 53, 54, 58, 69, 70, 78], as well as candidates for bariatric surgery [50, 68, 78], and adults of normal weight following a calorie restriction program [40]. The WEL was validated in Malay in a study by Singh *et al.* [81]. A short form has been created in English (WEL-SF [78]) which was later validated in Norwegian [74]. Apart from the Self-Regulation of Eating Attitude in Sports Scale of Scoffier *et al.* [66], there are currently no ESE scales in French, no French validation of the WEL, and no studies having explored the role of ESE among French speaking individuals with obesity. Furthermore, although some associations between self-efficacy and self-esteem, depression, well-being and stress have already been noted, additional research is needed to confirm and further explore these links. Thus, the present study was conducted to explore the relationship between eating self-efficacy and psychosocial correlates related to normal weight and obesity in French adults. Therefore, its aim was twofold. The first objective was to validate the French version of the scale and to explore its reliability, sensibility and measurement invariance in the general adult population and a clinical sample of overweight adults and those living with obesity. The second objective was to

explore the links between eating self-efficacy, sociodemographic variables, self-esteem, perceived stress, depression and well-being among these two samples.

## 2. Study 1

The aim of this study was to validate the French version of the WEL in the general adult population, explore its factorial structure, reliability and sensibility as well as the links with other psychosocial and sociodemographic variables.

### 2.1. Methods

**2.1.1. ESTEAM cohort.** ESTEAM is the acronym of "*Etude Scientifique des Tendances Empêchant un Amaigrissement Maintenu*" in French (translated in English as "Scientific Assessment of Tendencies Preventing Maintained Weight Loss"). It is a web-based retrospective study, funded by PiLeJe Laboratoire, a French pharmaceutical laboratory. Recruitment of the study population started in January 2014 and is still ongoing. This dynamic cohort contains data from psychological variables (such as depression, well-being, eating self-efficacy, perceived stress or self-esteem), collected from participants whose motivations to consult concern achieving and maintaining a healthy weight. Participants are recruited during a consultation with their doctor and on a voluntary basis. If the participant accepts to take part in the study, the doctor then creates his or her account. Participants can fill out the forms after each consultation with their practitioner, creating a follow-up database. Questionnaires can be completed directly online. Data storage is carried out by an independent host, certified by the French Ministry of Health and approved by French legislation. Each general practitioner is agreed by CNIL. Comité National Informatique et Liberté (National Comission on Informatics and Liberty in English) with a specific registration number.

When a new patient account is created, the physician asks for the patient's informed oral consent and checks the corresponding box on the software. The informed consent of the patient or their legal representative (parental authority for a minor, legal guardian for an adult under supervision) is mandatory for the creation of the patient account and the hosting of the collected data. The patient consent is dematerialized (stored in the computer system and not in the form of a paper document) and traced (the date of consent and the identity of the person who collects it are also kept). The extracted data are pseudoanonymized. Pseudonymisation is the processing of personal data carried out in such a way that data relating to an individual can no longer be attributed without additional information. Pseudonymization is one of the measures recommended by the General Data Protection Regulation (GDPR) to limit the risks associated with the processing of personal data (https://www.cnil.fr/fr/lanonymisation-de-donnees-personnelles): they contain the patient number.

**2.1.2. Participants.** The designated sample for the WEL validation analysis consisted of 432 participants (93% of whom were female). After removing participants whose gender or BMI data were missing in other scales, 374 participants (78% of whom were female) were included in the final sample intended for mean comparison and correlational analyses. For ethical reasons, participants were excluded if their age was under 18. Additionally, due to a low rate of participants over the age of 64, we decided to include those whose ages ranged from 18 to 64 (mean age 43.18 years, SD = 11.93). Moreover, participants were included if their BMI was less than 25 kg / m$^2$ (average BMI = 22.36 kg/m$^2$, SD = 2.16) and if they had completed the questionnaire only once in 21 days. Two hundred and one participants from the final sample filled out the depression questionnaire in addition to other scales.

**2.1.3. Measures.** *2.1.3.1. Eating self-efficacy*. Eating self-efficacy was assessed using the French version of the Clark, Abrams, Niaura, Eaton and Rossi's Weight Efficacy Life-Style

Questionnaire (WEL, 1991). The original items of the scale were adapted from Condiotte and Lichtenstein's Smoking Confidence Questionnaire (1981) [31] and some additional items were created from results of a clinical experience and the administration of the scale to a sample of patients with obesity. The original version of the WEL consists of 20 items, scored on a Likert-type scale (0: *Not confident*, 9: *Very confident*) and assembled into five different factors evaluating the individual's confidence of "being able to resist their desire to eat" [39] in the following situations: Negative emotions (items 1, 6, 11, 16), Availability (items 2, 7, 12, 17), Social Pressure (items 3, 8, 13, 18), Physical Discomfort (items 4, 9, 14, 19) and Positive Activities (items 5, 10, 15, 20). An important level of eating self-efficacy is characterized by a high score on the WEL. The values of Cronbach's alpha coefficient of all the scales used in this study are presented in Table 1.

*2.1.3.2. Well-being.* Well-being was evaluated with the WHO-5 Well Being Index, created and validated in French by the Psychiatric Research Unit WHO Collaborating Centre in Mental Health [82]. It is a unidimensional, 5-item questionnaire, evaluated using a Likert-type scale (0: *at no time*, 5: *all of the time*). A higher score on the scale indicates a higher level of well-being.

*2.1.3.3. Depressive symptomatology.* Depressive symptomatology was appraised using a French adaptation of the Major Depression Inventory (MDI), created by Psychiatric Research Unit [83] and specifically translated for the purpose of this study, following the usual scientific translation procedures [84]. It is a unidimensional scale consisting of 10 Likert-type items (0: *At no time*, 5: *All of the time*). This scale was administered only to participants who responded "At no time" or "Some of the time" on one of the five items of the WHO-5 scale or whose total WHO-5 score was inferior to 50. A high score on the MDI scale indicates a significant level of depressive symptomatology.

*2.1.3.4. State self-esteem.* State self-esteem was measured using the State Self-Esteem Scale (SSES) [85]. The original version of the SSES includes 20 items, scored on a Likert-type scale (1: *Not at all*; 5: *Extremely)* and grouped in three dimensions in order to assess performance self-esteem, social self-esteem and appearance self-esteem. In this research, a French version of the scale containing 12 items organized also in three dimensions was translated for the purpose of this study. A high score on this scale is a sign of high state self-esteem.

**Table 1. Cronbach's alpha coefficient of the scales used in studies 1 and 2.**

|  | General sample | Clinical sample |
|---|---|---|
|  | (study 1) | (study 2) |
| **Variables** | α | α |
| WEL | 0.91 | 0.91 |
| External stimuli | 0.91 | 0.88 |
| Internal stimuli | 0.90 | 0.90 |
| SSES | 0.84 | 0.85 |
| Performance | 0.81 | 0.83 |
| Social | 0.80 | 0.81 |
| Appearance | 0.85 | 0.84 |
| WHO-5 | 0.89 | 0.88 |
| MDI | 0.82 | 0.84 |
| PSS-11 | 0.87 | 0.87 |

*Note. WEL* = Weight Efficacy Life-Style Questionnaire; *SSES* = State Self-Esteem Scale; *WHO-5* = World Health Organisation Well-Being index; *MDI* = Major Depression Inventory; *PSS-11* = Perceived Stress Scale.

*2.1.3.5. Perceived stress.* Perceived stress was assessed with the Cohen, Kamarck and Mermelstein's Perceived Stress Scale (PSS) [86]. This unidimensional scale embodies 14 Likert-type items (1: *Never*, 5: *Very often*). A French version embodying 11 items was used in this study. A state of high stress is characterized by a high score on the scale.

*2.1.3.6. Anthropometric and demographic information.* In addition to the aforementioned questionnaires, the medical practitioner reported the participant's age, gender, weight and height, the latter two allowing their Body Mass Index (BMI) to be calculated.

**2.1.4. Statistical analysis.** In order to investigate the factorial structure of the WEL questionnaire, an exploratory factor analysis (EFA) followed by a confirmatory factor analysis (CFA) were performed. The data set was randomly split in half in order to carry out independent EFA and CFA analyses. Prior to the analysis, multivariate and univariate normalities of data were explored using the Mardia's coefficient and Shapiro-Wilk test, respectively. It was considered that data do not follow the normal distribution when the Mardia's coefficient $|z|$ score exceeds 5 and the Shapiro-Wilk test is significant ($p < 0.05$) [87] (p.32).

In addition, CFA was carried out on other scales used in the present study. These analyses confirmed the original structure of the WHO-5 and MDI while suggesting reduced versions of the SSES and the PSS in terms of number of retained items: the French version of the SSES scale contained 12 items instead of 20, and the French version of the PSS scale contained 11 instead of 14 items. However, the number of factors remained the same.

*2.1.4.1. Exploratory factor analysis (EFA).* The EFA was carried out on a first randomly split sample. Inter-item correlations as well as the Kaiser-Meyer-Olkin index (KMO) and the Bartlett sphericity index were calculated beforehand to determine if the data were suitable for an EFA. The significant values of the correlation coefficients and the Bartlett index (p < 0.05) as well as the satisfactory value of the KMO coefficient (KMO ≥ 0.70) indicated that the factor analysis could be performed as follows. The factor extraction method was chosen based on the normality of the distribution [87] (p.26). The number of factors retained was established using parallel analysis (eigenvalue > 1) and the Cattell criterion [88]. Finally, items that were not significantly correlated with other items of the scale, whose factor loadings were too low (< 0.40) or were loading on two or more factors were dismissed from the model.

*2.1.4.2. Confirmatory factor analysis (CFA).* Subsequently, CFA was conducted on the other split-half sample, testing the model derived from the EFA and using the lavaan package [89]. The goodness of fit of the model was analysed using the following indexes: $\chi2 / df$, Comparative Fit Index (CFI), Tucker-Lewis Index (TLI), Adjusted Goodness-of-Fit Index (AGFI), Root Mean Square Error of Approximation (RMSEA) and Standardized Root Mean Square (SRMR). Acceptable fit was considered when $\chi2 / df$ was inferior to 5, when CFI, TLI and AGFI values were superior to 0.90 and when RMSEA and SRMR values were less than 0.8 with values inferior to 0.6 being considered as excellent [90].

*2.1.4.3. Reliability and sensibility of the WEL.* The reliability of the WEL was studied by considering its internal consistency and using the split-half method. The internal consistency of the French version of the scale and its dimensions were analysed by calculating the Cronbach coefficient alpha (α). The interpretation of its value was based on the following criteria: $\alpha < 0.60$—unacceptable; $0.60 > \alpha > 0.70$—correct; $0.70 > \alpha > 0.80$—satisfactory; $0.80 > \alpha > 0.90$—very good; $\alpha > 0.95$—problematic [91]. Furthermore, the reliability of the scale was considered satisfactory if the interclass correlation coefficient (ICC), measured between even and odd items, was greater than 0.70 [92, 93]. Sensibility was investigated by performing discrimination analysis of items and by calculating the Ferguson's $\delta$ coefficient. This analysis relied on the following criteria: items with $d < 0.20$ are poor and need to be revised or eliminated; items with $d$ of 0.20–0.29 are marginal and need revision; items with $d$ of 0.30–0.39 are reasonably good but can be improved and items with $d > 0.40$ are very good [94]. Ferguson's $\delta$ coefficient

expresses the extent to which the scale discriminates between individuals and ranges from 0 to 1. A delta close to 0 indicates that all subjects have the same score while a delta close to 1 indicates that each subject has a different score [95].

*2.1.4.4. Relationships with sociodemographic and psychological variables.* The links between different variables were explored using nonparametric Spearman's *rhô* correlation coefficients. Non-parametric Kruskal-Wallis and Wilcoxon–Mann–Whitney U tests were conducted in order to identify any significant differences based on participants' gender, age and BMI. Cohen's *d* coefficient and $\eta^2$ index were applied in order to evaluate the size effect of those differences. According to Cohen [96], a size effect equal to 0.20 is considered small, a Cohen's *d* equal to 0.50 is considered as a medium effect and *d* = 0.80 is regarded as a large size effect. All the statistical analyses were performed using the R version 3.4.2. for Windows.

## 2.2. Results

Mean scores and standard deviations of the assessment scales used in the general French population sample are presented in Table 2.

**2.2.1. Exploratory Factor Analysis (EFA).** EFA was carried out on a first randomly split sample (*n* = 224). The absolute value of Mardia's coefficient | *z* | score was greater than 5 (| *z* | = 71.89, *p* = 0.00) and the Shapiro-Wilk test was significant (*p* < 0.001), indicating a defect in the multivariate and univariate normalities of the data. Therefore, nonparametric statistics were applied for the analyses performed on the French version of the WEL. The Bartlett's sphericity test and KMO were satisfactory (Bartlett test: $\chi^2$ (190) = 2972.932, *p* < 0.001; *KMO* = 0.94), confirming the suitability of the data for the EFA. The factor extraction method applied was the principal axis method since it does not require a normal distribution of the data [87] along with a varimax rotation, which was applied in the study by Clark *et al.* [39]. Analysis of the correlation matrix indicated that all items were significantly correlated with each other. Examination of the Cattell index and parallel analysis suggested a 2-factor solution. Items 2, 5, 7, 9, 10, 15, 17 and 20 were saturating on both factors and were hence excluded from the final model. The remaining 12 items explained 55% of the total variance. The EFA results are presented in Table 3.

**2.2.2. Confirmatory Factor Analysis (CFA).** CFA was carried out on the second randomly split sample (*n* = 208) using the MLR estimator. The first tested model was the 2-factor, 12-item solution. The results were the following: $\chi^2/df$ = 2.55; RSMEA = 0.086; SRMR = 0.072; AGFI = 0.848; CFI = 0.948; TLI = 0.935. Given the inadequate AGFI and RMSEA values, a second model was tested, incorporating the suggested modification indices. Hence, five correlated errors were added between items 3 and 18, items 11 and 19, items 1 and 6, items 1 and 3 and, lastly, items 12 and 18. These modifications significantly improved the overall fit of the model, according to the following Goodness of Fit Indices: $\chi^2/df$ = 1.67; RSMEA = 0.057; SRMR = 0.060; AGFI = 0.901; TLI = 0.972; CFI = 0.980. CFA results are displayed in Table 8. Considering the content of each subscale, the terms "External stimuli" (composed of items 3, 8, 12, 13 and 18) and "Internal stimuli" (items 1, 4, 6, 11, 14, 16 and 19) were given.

Due to the low proportion of underweight individuals and men in the general population sample, we were unable to perform multigroup confirmatory factor analysis to assess cross-BMI and cross-gender measurement invariance in this sample.

**2.2.3. Reliability and sensibility of the WEL.** The intra-class coefficient (ICC) with a Spearman-Brown correction was calculated between even and odd items and was equal to 0.92. Cronbach's alpha coefficient was 0.91 for the "External stimuli" dimension, 0.90 for the "Internal stimuli" dimension and 0.91 for the whole scale. Ferguson's $\delta$ coefficient was equal to

**Table 2. Means and standard deviations of the measured variables in a sample of the general French population.**

| Variables | M (SD) | Minimal | Maximal |
|---|---|---|---|
| WEL [a] | 71.17 (22.24) | 5 | 108 |
| External stimuli [a] | 30.79 (10.25) | 1 | 45 |
| Internal stimuli [a] | 40.37 (15.26) | 0 | 63 |
| SSES [a] | 46.45 (10.29) | 19 | 73 |
| Performance [a] | 25.21 (5.46) | 9 | 35 |
| Social [a] | 10.89 (3.74) | 4 | 20 |
| Appearance [a] | 10.35 (4.35) | 4 | 20 |
| WHO-5 [a] | 52.22 (22.75) | 0 | 100 |
| MDI [b] | 22.93 (8.48) | 5 | 50 |
| PSS-11 [a] | 31.39 (7.77) | 12 | 51 |
| Age [a] | 43.18 (11.93) | 18 | 64 |
| BMI [a] | 22.36 (2.16) | 13.3 | 24.99 |

*Note.*

[a] = Total sample (N = 374)

[b] = participants that responded to the MDI scale (n = 201); WEL = Weight Efficacy Life-Style Questionnaire; SSES = State Self-Esteem Scale; WHO-5 = World Health Organisation Well-Being index; MDI = Major Depression Inventory; PSS-11 = Perceived Stress Scale; BMI = Body Mass Index.

0.99 and discrimination analysis of the items was satisfactory, the discrimination index being superior to 0.40 for all items. These combined results support the hypothesis of adequate reliability and sensibility of the French version of the WEL.

**Table 3. Exploratory factor analysis results of the French version of the weight efficacy Life-Style questionnaire, carried out on the general population sample (n = 224).**

| WEL items | Factor loadings | |
|---|---|---|
| | F1 | F2 |
| 1. I can resist eating when I am anxious (nervous). | 0.8 | 0.24 |
| *2. I can control my eating on the weekends.* | *0.56* | *0.4* |
| 3. I can resist eating even when I have to say "no" to others. | 0.29 | 0.79 |
| 4. I can resist eating when I feel physically run down. | 0.65 | 0.23 |
| *5. I can resist eating when I am watching TV.* | *0.56* | *0.33* |
| 6. I can resist eating when I am depressed (or down). | 0.81 | 0.16 |
| *7. I can resist eating when there are many different kinds of foods available.* | *0.44* | *0.62* |
| 8. I can resist eating even when I feel it's impolite to refuse a second helping. | 0.2 | 0.84 |
| *9. I can resist eating even when I have a headache.* | *0.5* | *0.28* |
| *10. I can resist eating when I am reading.* | *0.47* | *0.26* |
| 11. I can resist eating when I am angry (or irritable). | 0.75 | 0.2 |
| 12. I can resist eating even when I am at a party. | 0.32 | 0.64 |
| 13. I can resist eating even when others are pressuring me to eat. | 0.17 | 0.83 |
| 14. I can resist eating when I am in pain. | 0.65 | 0.22 |
| *15. I can resist eating just before going to bed.* | *0.49* | *0.32* |
| 16. I can resist eating when I have experienced failure. | 0.84 | 0.19 |
| *17. I can resist eating even when high-calorie foods are available.* | *0.51* | *0.57* |
| 18. I can resist eating even when I think others will be upset if I don't eat. | 0.18 | 0.78 |
| 19. I can resist eating when I feel uncomfortable. | 0.73 | 0.16 |
| *20. I can resist eating when I am happy.* | *0.51* | *0.29* |

*Note.* Items in italics have been withdrawn from the French version used in study 1.

**Table 4. Correlations between measured variables: Results from study 1.**

| | 1. | 2. | 3. | 4. | 5. | 6. | 7. | 8. | 9. | 10. | 11. |
|---|---|---|---|---|---|---|---|---|---|---|---|
| 1. Age[a] | - | | | | | | | | | | |
| 2. BMI[a] | 0.19*** | - | | | | | | | | | |
| 3. WEL[a] | 0.05 | -0.19*** | - | | | | | | | | |
| 4.WEL-IS[a] | -0.01 | -0.15** | 0.81*** | - | | | | | | | |
| 5. WEL-ES[a] | 0.07 | -0.17** | 0.92*** | 0.50*** | - | | | | | | |
| 6. SSES[a] | 0.12* | -0.07 | 0.51*** | 0.42*** | 0.46*** | - | | | | | |
| 7. SSES-S[a] | 0.14** | 0 | 0.28*** | 0.27*** | 0.22*** | 0.63*** | - | | | | |
| 8. SSES-P[a] | -0.01 | 0.01 | 0.40*** | 0.33*** | 0.36*** | 0.84*** | 0.31*** | - | | | |
| 9. SSES-A[a] | 0.18*** | -0.17*** | 0.48*** | 0.35*** | 0.46*** | 0.76*** | 0.25*** | 0.47*** | - | | |
| 10. PSS-11[a] | -0.08 | 0.02 | -0.41*** | -0.29*** | -0.40*** | -0.60*** | -0.29*** | -0.53*** | -0.50*** | - | |
| 11.WHO-5[a] | 0.01 | -0.07 | 0.42*** | 0.30*** | 0.41*** | 0.57*** | 0.21*** | 0.51*** | 0.51*** | -0.64*** | - |
| 12. MDI[b] | 0.16* | 0.06 | -0.28*** | -0.15* | -0.30*** | -0.50*** | -0.23*** | -0.51*** | -0.31*** | 0.56*** | -0.62*** |

*Note.*

[a] = Total sample (*N* = 374)

[b] = participants who responded to the MDI (*n* = 201); *BMI* = Body Mass Index; *WEL* = Weight Efficacy Life-Style Questionnaire; *WEL–IS* = Internal Stimuli; *WEL–ES* = External Stimuli; *SSES* = State Self-Esteem Scale; *SSES–S* = State Self-Esteem–Social scale; *SSES–P* = State Self-Esteem–Performance scale; *SSES–A* = State Self-Esteem–Appearance scale; *PSS-11* = Perceived Stress Scale; *WHO-5* = World Health Organization Well-Being index; *MDI* = Major Depression Inventory.

\* $p<0.05$

\*\* $p<0.01$

\*\*\* $p<0.001$.

**2.2.4. Relationships with sociodemographic and psychological variables.** Analysis of the univariate and multivariate normalities indicated that the data from the SSES, PSS-11, WHO-5 and MDI questionnaires and their subscales were not distributed normally (Mardia's coefficient: $|z| > 5$, Shapiro normality test: $p < 0.001$). Therefore, the study of the links between the variables was carried out using nonparametric Spearman *rhô* correlation coefficients. Correlation coefficient values are presented in Table 4.

BMI displayed negative correlations with the total WEL score (*rhô* = -0.19) and its two subscales "Internal stimuli" (*rhô* = -0.15) and "External stimuli" (*rhô* = -0.17). The total WEL score and its two subscales were strongly correlated with each other. They also displayed identical correlation pattern with other variables. The total WEL score and the two subscales "Internal stimuli" and "External stimuli" were positively correlated with the total SSES score (*rhô* = 0.51; *rhô* = 0.42 and *rhô* = 0.46, respectively) and its subscales "Social (*rhô* = 0.28; *rhô* = 0.27 and *rhô* = 0.22, respectively), "Performance" (*rhô* = 0.40; *rhô* = 0.33 and *rhô* = 0.36, respectively) and "Appearance" (*rhô* = 0.48, *rhô* = 0.35 and *rhô* = 0.46, respectively). Additionally, they were positively correlated with the WHO-5 (*rhô* = 0.42; *rhô* = 0.30 and *rhô* = 0.41, respectively). On the other hand, negative correlations were observed between the total WEL score, "Internal stimuli", "External stimuli" subscales and PSS-11 (*rhô* = -0.41, *rhô* = -0.29 and *rhô* = -0.40, respectively) and MDI score (*rhô* = -0.28; *rhô* = -0.15 and *rhô* = -0.30, respectively).

Since data were not normally distributed, non-parametric Wilcoxon–Mann–Whitney U tests were used for mean comparison analyses. Significant differences in WEL scores and its two subscales were observed between the two BMI categories. Participants in the "Underweight" category had higher scores than those in the "Normal weight" category (*p* < 0.05). The size effect of those differences was respectively, *d* = -0.49, *d* = -0.45, and *d* = 0.14. Similarly, age seemed to be significantly different in the two BMI categories with participants in the "Underweight" category being younger than those in the "Normal weight" category (*p* < 0.001,

**Table 5. Means, standard deviations and Wilcoxon–Mann–Whitney statistics for study 1 variables.**

| | Female (n = 291) | Male (n = 83) | U (1, 372) | Underweight (n = 25) | Normal weight (n = 349) | U (1,372) | d |
|---|---|---|---|---|---|---|---|
| Measure | M (SD) | M (SD) | | M (SD) | M (SD) | | |
| BMI[a] | 22.40 (2.04) | 22.19 (2.55) | 12076 | 17.10 (1.31) | 22.73 (1.67) | 8725*** | 3.42 |
| Age[a] | 43.19 (11.76) | 43.17 (12.60) | 5965 | 34.52 (14.13) | 43.80 (11.53) | 60695*** | 0.79 |
| WEL[a] | 69.56 (22.47) | 84.37 (18.15) | 11194 | 81.44 (25.32) | 70.43 (21.86) | 2989** | -0.49 |
| WEL-IS[a] | 40.1 (15) | 41.34 (16.18) | 11397 | 46.76 (16.22) | 39.92 (15.11) | 3109.5* | -0.45 |
| WEL-ES[a] | 30.42 (10.37) | 32.12 (9.8) | 10897 | 34.68 (11.40) | 30.52 (10.13) | 3076* | 0.14 |
| SSES[a] | 46.6 (10.25) | 45.92 (10.45) | 12639 | 49.04 (10.01) | 46.27 (10.29) | 3732.5 | |
| SSES-S[a] | 10.96 (3.73) | 10.65 (3.79) | 12664 | 11.04 (3.81) | 10.88 (3.74) | 4334 | |
| SSES-P[a] | 25.21 (5.55) | 25.23 (5.15) | 12174 | 26.16 (5.33) | 25.14 (5.47) | 3847 | |
| SSES-A[a] | 10.44 (4.35) | 10.04 (4.38) | 1271 | 11.84 (4.09) | 10.24 (4.36) | 3400. | |
| PSS-11[a] | 31.29 (7.78) | 31.73 (7.74) | 11800 | 32.72 (7.47) | 31.30 (7.77) | 3881 | |
| WHO-5[a] | 51.77 (23.12) | 53.82 (21.43) | 11528 | 54.56 (18.54) | 52.06 (23.03) | 4132 | |
| | Female (n = 159) | Male (n = 42) | | Underweight (n = 11) | Normal weight (n = 190) | | |
| Variable | M (SD) | M (SD) | U (1,199) | M (SD) | M (SD) | U (1,199) | |
| MDI[b] | 23.06 (8.59) | 22.43 (8.14) | 23347 | 23.91 (5.11) | 22.87 (8.64) | 946 | |

*Note.*

[a] = Total sample (n = 374)

[b] = participants that responded to the MDI scale (n = 201); *WEL* = Weight Efficacy Life-Style Questionnaire; *WEL–IS* = Internal Stimuli; *WEL–ES* = External Stimuli; *SSES* = State Self-Esteem Scale; *SSES–S* = State Self-Esteem–Social scale; *SSES–P* = State Self-Esteem–Performance scale; *SSES–A* = State Self-Esteem–Appearance scale; *PSS-11* = Perceived Stress Scale; *WHO-5* = World Health organization Well-Being index; *MDI* = Major Depression Inventory; *BMI* = Body Mass Index. $p<0.1$

* $p<0.05$

** $p<0.01$

*** $p<0.001$.

$d = 0.79$). Results are displayed in Table 5. However, since 25 participants were included in the "Underweight" category, these results need to be confirmed in a larger sample.

### 2.3. Discussion

Results from EFA and CFA showed that the French 12-item-version of the WEL (WEL-Fr-G) is a psychometrically sound tool for assessing eating self-efficacy in a general population.

## 3. Study 2

The aim of study 2 was to validate the French version of the WEL in a clinical sample of French adults who were overweight or suffering from obesity as well as explore its psychometric properties (its factorial structure, reliability and sensibility) and its measurement invariance. This study also aimed to explore the links between eating self-efficacy and other psychological and sociodemographic variables.

### 3.1. Methods

**3.1.1. Participants.** Inclusion criteria for age were the same as in study 1 but inclusion criteria for BMI were different: participants with a BMI $\geq 25$ kg/m$^2$ were included. In total, 2010 adults (87% of whom were women) completed the French version of the WEL. Their responses were used for the validation analysis of the scale. After removing participants whose gender or

BMI data were missing from the other scales, 1737 participants (82% of whom were women) were included in the final sample for mean comparison and correlational analyses. Mean age was 44.44 years (SD = 11.25) and mean BMI was 32.29 kg/m$^2$ (SD = 5.62). A total of 967 participants in this sample completed the MDI questionnaire in addition to other scales.

**3.1.2. Procedure.** The same scales and statistical analyses as used in study 1 were applied in study 2 as well as an additional analysis of the measurement invariance of the WEL. To assess the measurement invariance of the scale across different BMI and gender categories, a series of multigroup confirmatory factor analyses (MGCFA) were performed using the lavaan [89] and semTools [97]. The measurement invariance was first tested across four different BMI categories: overweight group (BMI between 25.00 and 29.99 kg/m$^2$), category I obesity (BMI between 30.00 and 34.99 kg/m$^2$), category II obesity (BMI between 35.00 and 39.99 kg/m$^2$) and category III obesity (BMI above 40.00 kg/m$^2$). The measurement invariance was then tested between women and men. Four different models were tested and compared with each other. Model 1 tested *configural* invariance (testing of the hypothesized model across the different weight or gender categories, without constraints). Model 2 tested *metric* invariance (factor loadings were constrained to equality across weight or gender categories). Model 3 tested *scalar* invariance (factor loadings and item intercepts were constrained to equality between categories). Finally, model 4 tested *strict* invariance (factor loadings, item intercepts and item error variance were constrained to equality between categories). Measurement invariance of the models was supported if $\Delta\chi^2$ was not significant, ΔCFI ≤ 0.005, ΔRMSEA ≤ 0.01 and ΔSRMR ≤ 0.025 for the metric invariance test and if ΔCFI ≤ 0.005, ΔRMSEA ≤ 0.01 and ΔSRMR ≤ 0.005 for the scalar and strict invariance tests [98]. Before comparing the models, it was ensured that the configural model had an acceptable fit. For this purpose, the aforementioned CFA criteria were applied. After inspection of the configural model, measurement invariance was explored by comparing the four models with each other.

## 3.2. Results

Mean scores and standard deviations of the tools used in the second study are displayed in Table 6.

**3.2.1. Exploratory Factor Analysis (EFA).** EFA was carried out on a first randomly split sample. Forty-seven values were identified as outliers of the item 10 and were therefore excluded from the analysis. The final sample intended for the EFA included 924 participants. The absolute value of the Mardia's coefficient $z$ score was greater than 5 (| $z$ | = 123.441, $p$ = 0.00) and the Shapiro-Wilk test was significant ($p$ <0.001), hence nonparametric statistics were applied. The Bartlett's sphericity test and the Kaiser-Meyer-Olkin index (KMO) were satisfactory (Bartlett test: $\chi2$ (190) = 10132.75, $p$ < 0.001; $KMO$ = 0.94), indicating that data were suitable for EFA. Principal axis method was also used in this study as a factor extraction method. The correlation matrix indicated that all items were significantly correlated with each other. As in study 1, examination of the Cattell index and parallel analysis suggested a 2-factor solution. Items 2, 4, 5, 9, 10, 14, 15, 17 and 20 were saturating on both factors and were hence excluded from the factorial solution. The remaining 11 items accounted for 48% of the variance. The EFA results are presented in Table 7.

**3.2.2. Confirmatory Factor Analysis (CFA).** CFA was carried out on the second randomly split sample ($n$ = 1039) using MLR estimator. The first tested model was the 2-factor solution with 11 items. The results were: $\chi^2/df$ = 9.84; RSMEA = 0.091; SRMR = 0.046; AGFI = 0.895; CFI = 0.947; and TLI = 0.932. Given the inadequate values of some indexes (i.e. $\chi^2/df$, AGFI and RMSEA) a second model was tested, incorporating the suggested modification indices. Hence, two correlated errors between items 7 and 12 and items 3 and 18 were

**Table 6. Means and standard deviations of the assessment scales used in a French clinical sample.**

| Variables | M (SD) | Minimal | Maximal |
|---|---|---|---|
| WEL [a] | 58.64 (20.55) | 0 | 99 |
| Internal stimuli [a] | 24.37 (12.01) | 0 | 45 |
| External stimuli [a] | 34.27 (11.51) | 0 | 54 |
| SSES [a] | 44.12 (9.53) | 15 | 73 |
| Performance [a] | 25.13 (5.30) | 7 | 35 |
| Social [a] | 10.97 (3.77) | 4 | 20 |
| Appearance [a] | 8.02 (3.93) | 4 | 20 |
| WHO-5 [a] | 51.45 (22.96) | 0 | 100 |
| MDI [b] | 23.4 (8.77) | 2 | 49 |
| PSS-11 [a] | 31.54 (7.87) | 11 | 55 |
| Age [a] | 44.44 (11.25) | 18 | 64 |
| BMI [a] | 32.29 (5.62) | 30 | 67.5 |

*Note.*

[a] = Total sample (N = 1737)

[b] = participants that responded to the MDI scale (n = 967); *WEL* = Weight Efficacy Life-Style Questionnaire; *SSES* = State Self-Esteem Scale; *WHO-5* = World Health Organisation Well-Being index; *MDI* = Major Depression Inventory; *PSS-11* = Perceived Stress Scale; *BMI* = Body Mass Index.

added, significantly improving the overall fit of the model. The results from this second and final analysis were: $\chi^2/df$ = 5.56; RSMEA = 0.066; SRMR = 0.040; AGFI = 0.938; TLI = 0.964; CFI = 0.973. CFA results are presented in Table 8. Given the content of each subscale, names

**Table 7. Exploratory factor analysis results of the French version of the weight efficacy Life-Style questionnaire, carried out on the clinical sample (n = 924).**

| | Factor loading | |
|---|---|---|
| **WEL items** | **F1** | **F2** |
| 1. I can resist eating when I am anxious (nervous). | 0.29 | 0.77 |
| *2. I can control my eating on the weekends.* | *0.58* | *0.38* |
| 3. I can resist eating even when I have to say "no" to others. | 0.76 | 0.28 |
| *4. I can resist eating when I feel physically run down.* | *0.32* | *0.53* |
| *5. I can resist eating when I am watching TV.* | *0.35* | *0.33* |
| 6. I can resist eating when I am depressed (or down). | 0.21 | 0.83 |
| 7. I can resist eating when there are many different kinds of foods available. | 0.63 | 0.30 |
| 8. I can resist eating even when I feel it's impolite to refuse a second helping. | 0.75 | 0.14 |
| *9. I can resist eating even when I have a headache.* | *0.43* | *0.33* |
| *10. I can resist eating when I am reading.* | *0.41* | *0.24* |
| 11. I can resist eating when I am angry (or irritable). | 0.23 | 0.70 |
| 12. I can resist eating even when I am at a party. | 0.58 | 0.20 |
| 13. I can resist eating even when others are pressuring me to eat. | 0.78 | 0.18 |
| *14. I can resist eating when I am in pain.* | *0.37* | *0.42* |
| *15. I can resist eating just before going to bed.* | *0.35* | *0.32* |
| 16. I can resist eating when I have experienced failure. | 0.25 | 0.80 |
| *17. I can resist eating even when high-calorie foods are available.* | *0.64* | *0.36* |
| 18. I can resist eating even when I think others will be upset if I don't eat. | 0.73 | 0.21 |
| 19. I can resist eating when I feel uncomfortable. | 0.24 | 0.71 |
| *20. I can resist eating when I am happy.* | *0.48* | *0.22* |

*Note.* Items in italics have been withdrawn from the version used in study 2.

**Table 8. Goodness of fit indices of the two weight efficacy Life-Style questionnaire versions used in study 1 and study 2.**

| Study | $\chi^2$ | $p$ | $\chi^2/df$ | RMSEA | SRMR | AGFI | TLI | CFI |
|-------|-------|------|-------------|-------|-------|-------|-------|-------|
| Study 1 | 80.2 | 0.00 | 1.67 | 0.057 | 0.06 | 0.901 | 0.972 | 0.980 |
| Study 2 | 227.963 | 0.00 | 5.56 | 0.066 | 0.040 | 0.938 | 0.964 | 0.973 |

*Note.* RMSEA = Root Mean Square Approximation; SRMR = Standardized Root Mean Square; AGFI = Adjusted Goodness-of -Fit Index; TLI = Tucker-Lewis Index; CFI = Comparative Fit Index.

identical to those in study 1 were given to the subscales: "External stimuli" (composed of items 3, 7, 8, 12, 13, and 18) and "Internal stimuli" (including items 1, 6, 11, 16 and 19).

**3.2.3. Multigroup confirmatory factor analysis (MGCFA).** A series of MGCFA was conducted to test for cross-BMI and cross-gender measurement invariance.

*3.2.3.1. Measurement invariance across BMI categories.* For this analysis, the 2010 participants were divided into four categories according to their BMI. The overweight category included 841 participants (91% female; mean age = 45.18, SD = 11.32, age range: 18–64 years; mean BMI = 27.49 kg/m$^2$, SD = 1.43). Category I obesity was composed of 661 participants (84% female; mean age = 44.38, SD = 11.42, age range: 18–64 years; mean BMI = 32.23 kg/m$^2$, SD = 1.47). Category II obesity included 306 participants (85% female; mean age = 43.78, SD = 11.34, age range: 18–64; mean BMI = 37.2 kg/m$^2$, SD = 1.63) and category III obesity had 202 participants (85% female; mean age = 44.1, SD = 10.86, age range: 19–64 years; mean BMI = 44.14 kg/m$^2$, SD = 4.13).

First, a regular CFA was carried out separately in each of the four BMI categories.

The fit indices of the model tested in the four groups were good, as presented in Table 9.

Overweight: CFI = 0.974; TLI = 0.965; SRMR = 0.043; RMSEA = 0.065 (90% CI 0.053–0.076);

Category I: CFI = 0.982; TLI = 0.975; SRMR = 0.043; RMSEA = 0.056 (90% CI 0.043–0.069);

Category II: CFI = 0.987; TLI = 0.983; SRMR = 0.041; RMSEA = 0.048 (90% CI 0.020–0.071);

Category III: CFI = 0.984; TLI = 0.979; SRMR = 0.051; RMSEA = 0.049 (90% CI 0.020–0.080).

Subsequently, four different models were explored, testing respectively for configural, metric, scalar and strict factorial invariance. Comparison of the configural and metric measurement invariance showed that the added constraint of factor loadings did not significantly weaken the goodness of fit ($\Delta$CFI = 0.001; $\Delta$SRMR = 0.005; $\Delta$RMSEA = 0.003; $\Delta\chi^2$ = 47.33, $p$ = 0.03). Then, comparison of model 3 and model 2 revealed that the additional constraints requiring items' intercept equality did not significantly worsen the fit ($\Delta$CFI = 0.001; $\Delta$SRMR = 0.001; $\Delta$RMSEA = 0.003; $\Delta\chi^2$ = 38.67). Lastly, comparison of model 4 and model 3 pointed out that the further constraint expecting item error variances to be equal did not significantly affect the fit ($\Delta$CFI = 0.001; $\Delta$SRMR = 0.001; $\Delta$RMSEA = 0.002; $\Delta\chi^2$ = 38.52). These results support the idea of WEL invariance according to the BMI.

*3.2.3.2. Measurement invariance across gender.* A MGCFA analysis was run to examine the measurement invariance of the French version of the WEL across gender. For this analysis, the 2010 participants were divided according to their gender, i.e. into a group of 1747 women (mean age = 44.37, SD = 11.31, age range: 18–64 years) and a group of 263 men (mean age = 46.08, SD = 11.25, age range: 18–64 years). Just like in the previous analysis, a regular CFA was conducted separately in each of the two groups.

**Table 9. Goodness of fit indices of models testing for Cross-BMI and Cross-gender invariance in study 2.**

|  | CFI | SRMR | RMSEA | $\chi 2$ | ΔCFI | ΔSRMR | ΔRMSEA | Δ $\chi$ 2 |
|---|---|---|---|---|---|---|---|---|
| BMI category |  |  |  |  |  |  |  |  |
| Overweight | 0.974 | 0.043 | 0.065 | 146.183 |  |  |  |  |
| Category I obesity | 0.982 | 0.043 | 0.056 | 105.143 |  |  |  |  |
| Category II obesity | 0.987 | 0.041 | 0.048 | 61.687 |  |  |  |  |
| Category III obesity | 0.984 | 0.051 | 0.049 | 54.980 |  |  |  |  |
| Gender group |  |  |  |  |  |  |  |  |
| Female | 0.977 | 0.040 | 0.061 | 235.085 |  |  |  |  |
| Male | 0.987 | 0.039 | 0.047 | 58.644 |  |  |  |  |
| Cross-BMI invariance |  |  |  |  |  |  |  |  |
| Configural | 0.980 | 0.044 | 0.058 | 365.663 |  |  |  |  |
| Metric | 0.979 | 0.049 | 0.055 | 412.994 | 0.001 | 0.005 | 0.003 | 47.33* |
| Scalar | 0.978 | 0.050 | 0.052 | 451.663 | 0.001 | 0.001 | 0.003 | 38.67 |
| Strict | 0.977 | 0.049 | 0.050 | 490.183 | 0.001 | 0.001 | 0.002 | 38.52 |
| Cross-gender invariance |  |  |  |  |  |  |  |  |
| Configural | 0.978 | 0.040 | 0.060 | 296.083 |  |  |  |  |
| Metric | 0.978 | 0.042 | 0.057 | 312.749 | 0.000 | 0.002 | 0.003 | 16.67 |
| Scalar | 0.976 | 0.043 | 0.056 | 342.899 | 0.002 | 0.001 | 0.001 | 30.15*** |
| Strict | 0.976 | 0.044 | 0.053 | 349.69 | 0.000 | 0.001 | 0.003 | 6.79 |

*Note.* CFI = Comparative Fit Index; *RMSEA* = Root Mean Square Approximation; *SRMR* = Standardized Root Mean Square.

* $p<0.05$

** $p<0.01$

***$p<0.001$.

The fit indices of the model tested across the two gender groups were satisfactory: in the group of women: CFI = 0.977; TLI = 0.969; SRMR = 0.040; RMSEA = 0.061 (90% CI 0.054–0.069), and in the group of men: CFI = 0.987; TLI = 0.982; SRMR = 0.039; RMSEA = 0.047 (90% CI 0.012–0.073). Next, four different models of measurement invariance were explored. The results are presented in Table 9. The comparison of the configural and metric measurement invariance indicated that the added constraint did not significantly lessen the goodness of fit (ΔCFI = 0.000; ΔSRMR = 0.002; ΔRMSEA = 0.003; $\Delta\chi^2$ = 16.67). Second, the comparison between models 3 and 2 showed that the additional constraint did not significantly weaken the fit (ΔCFI = 0.002; ΔSRMR = 0.001; ΔRMSEA = 0.001; $\Delta\chi^2$ = 30.15, $p < 0.001$). Lastly, the comparison of models 4 and 3 also revealed that the added constraint did not significantly alter the fit (ΔCFI = 0.000; ΔSRMR = 0.001; ΔRMSEA = 0.003; $\Delta\chi^2$ = 6.79). These results support the hypothesis of cross-gender measurement invariance of the WEL.

**3.2.4. Reliability and sensibility of the WEL.** The intra-class coefficient (ICC), calculated between the even and odd items, with a Spearman-Brown correction, was equal to 0.87. Cronbach's alpha coefficient was 0.88 for the "External stimuli" dimension, 0.90 for the "Internal stimuli" dimension and 0.91 for the whole scale. These indices indicate a satisfactory internal consistency of the scale. A good sensibility of the scale was supported by a satisfactory Ferguson's delta coefficient ($\delta$ = 0.99) and a good discrimination analysis of the items with a discrimination index superior to 0.40 for all the items. In conclusion, these results endorse satisfactory reliability and sensibility of the French validation of the WEL.

**3.2.5. Relationships with sociodemographic and psychological variables.** Given the fact that the data were not distributed normally (Mardia coefficient: $|z| > 5$, Shapiro normality test: $p < 0.001$), nonparametric Spearman *rhô* correlation coefficients were used in order to

**Table 10. Correlations between measured variables: Results from study 2.**

| | 1. | 2. | 3. | 4. | 5. | 6. | 7. | 8. | 9. | 10. | 11. |
|---|---|---|---|---|---|---|---|---|---|---|---|
| 1. Age[a] | - | | | | | | | | | | |
| 2. BMI[a] | -0.05 | - | | | | | | | | | |
| 3. WEL[a] | 0.05* | -0.05 | - | | | | | | | | |
| 4. WEL-IS[a] | 0.06** | -0.08** | 0.88*** | - | | | | | | | |
| 5. WEL-ES[a] | 0.03 | 0 | 0.87*** | 0.53*** | - | | | | | | |
| 6. SSES[a] | 0.12*** | -0.02 | 0.47*** | 0.44*** | 0.38*** | - | | | | | |
| 7. SSES-S[a] | 0.22*** | 0 | 0.23*** | 0.22*** | 0.18*** | 0.64*** | - | | | | |
| 8. SSES-P[a] | -0.02 | 0.04 | 0.40*** | 0.36*** | 0.34*** | 0.79*** | 0.21*** | - | | | |
| 9. SSES-A[a] | 0.1*** | -0.11*** | 0.38*** | 0.38*** | 0.28*** | 0.74*** | 0.30*** | 0.38*** | - | | |
| 10. PSS-11[a] | -0.06** | 0.02 | -0.46*** | -0.46*** | -0.35*** | -0.56*** | -0.27*** | -0.55*** | -0.37*** | - | |
| 11.WHO-5[a] | 0.03 | -0.02 | 0.40*** | 0.41*** | 0.29*** | 0.53*** | 0.18*** | 0.52*** | 0.42*** | -0.65*** | - |
| 12. MDI[b] | 0.05 | -0.02 | -0.28*** | -0.26*** | -0.22*** | -0.51*** | -0.22*** | -0.52*** | -0.29*** | 0.55*** | -0.64*** |

*Note.*

[a] = Total sample ($N$ = 1737)

[b] = participants who responded to the MDI ($n$ = 967); BMI = Body Mass Index; WEL = Weight Efficacy Life-Style Questionnaire; WEL–IS = Internal Stimuli; WEL–ES = External Stimuli; SSES = State Self-Esteem Scale; SSES–S = State Self-Esteem–Social scale; SSES–P = State Self-Esteem–Performance scale; SSES–A = State Self-Esteem–Appearance scale; PSS-11 = Perceived Stress Scale; WHO-5 = World Health Organization Well-Being index; MDI = Major Depression Inventory.

* $p < 0.05$

** $p < 0.01$

*** $p < 0.001$.

examine the links between the variables. The correlation coefficient values are displayed in Table 10.

Age was positively but weakly correlated with the total WEL score ($rhô$ = 0.05) and its "Internal stimuli" subscale ($rhô$ = 0.06). BMI was negatively correlated with the "Internal stimuli" subscale only ($rhô$ = -0.08). The total WEL score and its two subscales were strongly correlated with each other. They also displayed identical correlation pattern with other variables. The total WEL score and the two subscales were positively correlated with the total SSES score ($rhô$ = 0.47; $rhô$ = 0.44 and $rhô$ = 0.38, respectively) and its three sub-scales "Social" ($rhô$ = 0.23; $rhô$ = 0.22 and $rhô$ = 0.18, respectively), "Performance" ($rhô$ = 0.40; $rhô$ = 0.36 and $rhô$ = 0.34, respectively) and "Appearance" ($rhô$ = 0.38 and $rhô$ = 0.28). Additionally, they were positively correlated to the WHO-5 scale ($rhô$ = 0.40; $rhô$ = 0.41 and $rhô$ = 0.29, respectively). On the other hand, negative correlations were observed between the total WEL score, "Internal stimuli" and "External stimuli" subscales and the PSS-11 ($rhô$ = -0.46 and $rhô$ = -0.35) and MDI scores ($rhô$ = -0.28; $rhô$ = -0.26 and $rhô$ = -0.22, respectively).

Since the data were not distributed normally, non-parametric Kruskall-Wallis and Wilcoxon–Mann–Whitney U tests were used for mean comparison analyses. The results are displayed in Table 11. Significant differences of WEL scores were observed between the four BMI categories. Participants in Category III obesity had the lowest WEL scores when compared to Overweight participants or Category I obesity ($p$ = 0.026). However, the size effect of these differences was extremely small ($\eta^2$ = 0.005). Likewise, when compared to the other three BMI categories, participants in category III obesity had the lowest eating self-efficacy towards internal stimuli ($p < 0.05$) but the size effect of these differences was also very small ($\eta^2$ = 0.008).

**Table 11. Means, standard deviations, Wilcoxon–Mann–Whitney and Kruskall-Wallis statistics for study 2 variables.**

|  | Female | Male |  | Overweight | Category I obesity | Category II | Category III obesity |  |  |
|---|---|---|---|---|---|---|---|---|---|
|  | (*n* = 1427) | (*n* = 310) |  | (*n* = 710) | (*n* = 581) | obesity (*n* = 265) | (*n* = 181) |  |  |
| Variable | M (SD) | M (SD) | U (1, 1735) | M (SD) | M (SD) | M (SD) | M (SD) | $\chi^2$ (3) | $\eta^2$ |
| BMI[a] | 32.26 (5.63) | 32.47 (5.59) | 9199 | 27.50 (1.42) | 32.21 (1.45) | 37.16 (1.37) | 44.25 (1.33) | 1544.4*** | 0.88 |
| Age[a] | 44.4 (11.26) | 44.64 (11.19) | 217560 | 45.20 (11.11) | 44.00 (11.48) | 43.71 (11.31) | 43.97 (10.84) | 5.415 |  |
| WEL[a] | 58.49 (20.64) | 59.33 (20.15) | 217556 | 59.00 (20.59) | 59.23 (20.18) | 59.26 (21.15) | 54.39 (20.51) | 9.27* | 0.005 |
| WEL–IS[a] | 24.27 (12.04) | 24.84 (11.89) | 215583 | 24.97 (11.87) | 24.59 (12.00) | 24.32 (12.03) | 21.36 (12.23) | 13.066** | 0.008 |
| WEL–ES[a] | 34.22 (11.62) | 34.49 (11.02) | 219442 | 34.03 (11.66) | 34.64 (11.14) | 34.94 (11.85) | 33.03 (11.61) | 3.66 |  |
| SSES[a] | 43.49 (9.53) | 44.95 (9.49) | 206266. | 44.03 (9.46) | 44.49 (9.71) | 44.39 (9.52) | 42.92 (9.21) | 1.334 [c] |  |
| SSES–S[a] | 10.91 (3.79) | 11.21 (3.70) | 209931 | 10.84 (3.85) | 11.17 (3.72) | 10.90 (3.67) | 10.91 (3.79) | 2.96 |  |
| SSES–P[a] | 25.08 (5.28) | 25.39 (5.36) | 213725 | 24.79 (5.26) | 25.37 (5.28) | 25.68 (5.20) | 24.96 (5.56) | 8.05. |  |
| SSES–A[a] | 7.95 (3.92) | 8.35 (3.95) | 206888. | 8.41 (4.02) | 7.95 (3.89) | 7.81 (4.00) | 7.06 (3.36) | 20.21*** | 0.01 |
| PSS-11[a] | 31.65 (7.87) | 31.04 (7.82) | 230984 | 31.96 (7.82) | 30.86 (7.81) | 30.90 (7.60) | 32.99 (8.34) | 13.149** | 0.008 |
| WHO-5[a] | 51.24 (23.06) | 53.39 (22.55) | 215230 | 50.81 (23.29) | 52.53 (22.43) | 52.51 (22.55) | 48.91 (23.84) | 4.94 |  |
|  | Female | Male |  | Overweight | Category I obesity | Category II obesity | Category III obesity |  |  |
|  | (*n* = 797) | (n = 170) |  | (*n* = 410) | (*n* = 313) | (*n* = 136) | (*n* = 108) |  |  |
| Variable | M (SD) | M (SD) | U (1,965) | M (SD) | M (SD) | M (SD) | M (SD) | $\chi^2$ (3) | $\eta^2$ |
| MDI[b] | 23.56 (8.85) | 22.69 (8.36) | 71550 | 13.82 (13.59) | 12.23 (13.11) | 11.65 (12.89) | 14.51 (13.51) | 9.098** | 0.006 |

*Note.*

[a] = Total sample (*n* = 1737*)*

[b] = participants that responded to the MDI scale (*n* = 967); [c] = result obtained with ANOVA (*F*(3, 1733)), because the SSES distribution was normal; *WEL* = Weight Efficacy Life-Style Questionnaire; *WEL–IS* = Internal Stimuli; *WEL–ES* = External Stimuli; *SSES* = State Self-Esteem Scale; *SSES–S* = State Self-Esteem–Social scale; *SSES–P* = State Self-Esteem–Performance scale; *SSES–A* = State Self-Esteem–Appearance scale; *PSS-11* = Perceived Stress Scale; *WHO-5* = World Health Organization Well-Being index; *MDI* = Major Depression Inventory; *BMI* = Body Mass Index. *p*<0.1

* *p*<005

** *p*<0.01

*** *p*<0.001.

## 3.3. Discussion

Our findings indicate that the French 11-item version of the WEL (WEL-Fr-C) can be administered to a clinical population of adults suffering from obesity, regardless of their BMI. Good psychometric properties and the established measurement invariance of the WEL-Fr-C endorse the idea that it is a valid instrument for assessing self-efficacy for controlled eating in both men and women from a clinical group. Results from mean comparisons and correlation analyses help improve the description of characteristics and difficulties present in a clinical population and related to eating self-efficacy.

## 4. Discussion

The aim of this research was to validate the French version of the Weight Efficacy Life-Style questionnaire in general population and clinical samples, to explore the psychometric properties of the scale, its measurement invariance as well as the relationships with other variables included in the two studies. Exploratory and confirmatory factor analyses yielded a two-factor structure of the scale for each of the samples, but provided two different French versions of WEL with specific items: 12 items in the general population sample (items: 1, 3, 4, 6, 8, 11, 12, 13, 14, 16, 18 and 19; WEL-Fr-G) and 11 items in the clinical sample (items: 1, 3, 6, 7, 8, 11, 12, 13, 16, 18 and 19; WEL-Fr-C). Compared to the original scale [39], which embodies 20 items

and three factors, the factorial structure and the number of items are different in the two French versions. For WEL-Fr-G, items belonging to the Positive Activities subscale from the original version (items 5, 10, 15 and 20) were completely removed, along with items 2, 7 and 17 from the Availability scale and item 9 from the Physical Discomfort scale. For WEL-Fr-C, the same items were removed with the exception of item 7, which was kept and items 4 and 14 (from the original Physical Discomfort scale), which were removed. The English WEL-SF [78] and its Norwegian version [74] share several removed items with the French versions: items 9, 10, 15, 17 and 20. However, no statistical parameter can explain the removal of items 2, 4 and 5 from the French versions. Therefore, their withdrawal may be due to the ambiguity of their content in French.

The reliability analysis of the two versions of the French WEL validation revealed a strong internal consistency, according to the Cronbach's alpha coefficients and ICC. For the WEL-Fr-G, $\alpha = 0.91$ for the general score and ICC = 0.92; $\alpha = 0.90$ for the "External stimuli" scale and $\alpha = 0.91$ for the "Internal stimuli" scale. For the WEL-Fr-C, $\alpha = 0.91$ for the general score and ICC = 0.86; $\alpha = 0.88$ for the "External stimuli" scale and $\alpha = 0.90$ for the "Internal stimuli" scale. These values are similar to those declared in other validations and adaptations of the WEL, with alpha values ranging from 0.78 to 0.92 [39, 74, 78, 81]. In addition to a strong reliability, the French version of the scale presented satisfactory sensibility as well. Moreover, measurement invariance analysis revealed that WEL-Fr-C can be used in a clinical sample regardless of participants' gender or BMI.

Body Mass Index was negatively correlated with the general score of WEL-Fr-G and WEL-Fr-C and the two subscales of WEL-Fr-G. For WEL-Fr-C, only "Internal stimuli" scale displayed significant negative correlation with BMI. These results are in line with previous studies which reported negative correlations between ESE and BMI [22, 40, 47, 51, 63, 65, 68, 72, 73]. Contrary to the results presented by Ames, Heckman, Diehl, Grothe and Clark [68], age was positively correlated with the general score of WEL-Fr and the "Internal stimuli" subscale of WEL-Fr-C. In both the general and the clinical samples, ESE was positively correlated with the self-esteem and well-being and negatively correlated with perceived stress and depressive symptoms. These results corroborate those from previous studies [55, 62, 63, 67–69]. However, a few differences were observed when comparing the correlation coefficients of the two subscales of the French WEL and the other variables in the two samples: for WEL-Fr-G, the "External stimuli" subscale displayed stronger links with Appearance self-esteem, well-being and perceived stress than the "Internal stimuli" subscale. Conversely, for WEL-Fr-C, the "Internal stimuli" subscale displayed stronger correlations with Appearance self-esteem, well-being and perceived stress than the "External stimuli" subscale. We can suppose that, in a general population, eating self-efficacy when faced with external stimuli is more predictive of impairment and a state of well-being than self-efficacy in the context of internal stimuli. Accordingly, in a clinical sample, eating self-efficacy in the context of internal stimuli may be more predictive of impairment and a state of well-being, than self-efficacy in the context of external stimuli. Significant differences in eating self-efficacy tendencies were observed in both samples according to the BMI category. Participants in the "Underweight" BMI category had higher eating self-efficacy scores than those in the "Normal weight" category. Likewise, participants in "Category III obesity" reported the lowest levels of eating self-efficacy when compared to the other three BMI categories. Similar differences were reported in a study by Menéndez-González and Orts-Cortés [72]. However, as in the studies by Ames *et al.* [68] and Clark and King [58], no significant differences were observed in eating self-efficacy trends by gender in either sample.

This research carries some limits. Because of uneven and small samples of men and underweight participants, mean comparison tests as well as cross-BMI and cross-gender

measurement invariance will need to be confirmed in larger general population samples. Also, the cross-sectional design of the study made it impossible to explore the test-retest reliability of the two versions of the French WEL scale as well as to identify the predictors of ESE. Future studies using a longitudinal design are needed to identify these predictors. Given the fact that eating self-efficacy plays an important role in various health behaviours and in different process related to weight loss and its maintenance (adherence to a new behaviour, early withdrawal, relapse, etc.), future studies could focus on creating comprehensive and predictive models of weight loss and weight maintenance incorporating eating and weight related self-efficacy. Moreover, only one study explored a link between ESE and calorie restriction diet in a general population [40]. It could be interesting to explore the nature of the links between ESE and intermittent fasting in a general population, not seeking to lose weight. Furthermore, some studies explored the links between ESE and eating disorders. They reported negative correlations between the WEL score and binge eating disorder [64, 69, 99] as well as some symptoms of eating disorders as measured by the EDI-2 [64] and EAT-26 [65]. Links with ESE, eating disorders and different eating styles remain to be investigated. Moreover, to our knowledge, only one study explored the links between orthorexia nervosa and general self-efficacy [100]. The authors found that the cluster of participants that had the healthiest diet also had the greatest general self-efficacy scores. Those participants also had higher orthorexic tendencies, which are described as the pathological obsession for healthy food [101]. Therefore, the following studies could also explore the negative consequences of high levels of eating-self efficacy, especially in the context of orthorexia nervosa.

The use of the French version of the WEL can contribute to a better evaluation of eating behaviours in diverse populations, both general and clinical. The application of the two versions of the scale is not exclusive to mental health professionals: it can be used by physicians with patients whose treatment includes diet adjustments. The results of our study highlight the important role that psychological factors play in individuals with normal weight, but also in patients with obesity. Multidisciplinary interventions have been shown to be effective for weight loss or a disease management that includes diet adjustment. Therefore, nutritional interventions should take into account the psychological factors related to eating behaviour. Although we emphasize multidisciplinary interventions on weight loss in adults with obesity by taking into account biological, psychological, and social factors, we are aware that such actions are sometimes difficult to implement. Too often, obesity treatment focus on dietary changes and the promotion of physical activity. In addition, research on psychological factors do not systematically examine general and eating self-efficacy.

## 5. Conclusions

Based on the results from this research, two different versions of the WEL should be used: the WEL-Fr-G for the assessment of ESE in the general French population and the WEL-Fr-C for a use in a clinical population. The results from this research suggest that the two French versions of the WEL are psychometrically sound instruments for assessing self-efficacy for controlled eating. Further studies will be necessary to explore eating self-efficacy in different contexts and different populations (individuals suffering from eating disorders, chronic diseases, individuals who are fasting intermittently etc.) and its links with specific eating behaviours or disorders.

## Acknowledgments

The authors would like to give special thanks to Michel Dubourdeaux (R&D Innovation Director at PiLeJe), Christophe Guyomar (Chief Digital Officer at PiLeJe) for their administrative

and technical support and Claude Blondeau (Head of Scientific Communication at PiLeJe) for editing the manuscript. Thanks are also addressed to Dr Dominique Périn Calvão for the translation of the MDI, Dr Marilou Bruchon-Schweitzer, Dr Eve Villemur and Dr Gérard Ostermann for their contribution to the translation of the WEL and SSES scales.

## Author Contributions

**Conceptualization:** Natalija Plasonja, Anna Brytek-Matera, Greg Décamps.

**Formal analysis:** Natalija Plasonja.

**Funding acquisition:** Natalija Plasonja, Greg Décamps.

**Methodology:** Natalija Plasonja, Anna Brytek-Matera, Greg Décamps.

**Project administration:** Greg Décamps.

**Resources:** Natalija Plasonja.

**Supervision:** Greg Décamps.

**Validation:** Natalija Plasonja, Anna Brytek-Matera, Greg Décamps.

**Writing – original draft:** Natalija Plasonja.

**Writing – review & editing:** Natalija Plasonja, Anna Brytek-Matera, Greg Décamps.

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
