## [Decision Letter · Decision Letter 0]

26 Jul 2021

PONE-D-21-08991

French validation of the Weight Efficacy Life-Style questionnaire (WEL): Links with mood, self-esteem and stress among the general population and a clinical sample of individuals with overweight and obesity

PLOS ONE

Dear Dr. Plasonja,

Thank you for submitting your manuscript to PLOS ONE. After careful consideration, we feel that it has merit but does not fully meet PLOS ONE’s publication criteria as it currently stands. Therefore, we invite you to submit a revised version of the manuscript that addresses the points raised during the review process.

We look forward to receiving your revised manuscript.

Kind regards,

Antonio Palazón-Bru, PhD

Academic Editor

PLOS ONE

Journal Requirements:

4. Please provide additional details regarding participant consent. In the ethics statement in the Methods and online submission information, please ensure that you have specified (1) whether consent was informed and (2) what type you obtained (for instance, written or verbal, and if verbal, how it was documented and witnessed). If your study included minors, state whether you obtained consent from parents or guardians. If the need for consent was waived by the ethics committee, please include this information.

The authors would like to give special thanks to Michel Dubourdeaux (R&D Innovation Director at PiLeJe), Christophe Guyomar (Chief Digital Officer at PiLeJe) for their administrative and technical support and Claude Blondeau (Head of Scientific Communication at PiLeJe) for editing the manuscript. Thanks are also addressed to Dr Dominique Périn Calvão for the translation of the MDI, Dr Marilou Bruchon-Schweitzer, Dr Eve Villemur and Dr Gérard Ostermann for their contribution to the translation of the WEL and SSES scales. Anna Brytek-Matera acknowledges financial support by the „Excellence Initiative – Research University” program for years 2020-2026 of the University of Wroclaw.

Anna Brytek-Matera acknowledges financial support by the „Excellence Initiative – Research University” program for years 2020-2026 of the University of Wroclaw.

Reviewers' comments:

Reviewer's Responses to Questions

**Comments to the Author**

1. Is the manuscript technically sound, and do the data support the conclusions?

Reviewer #1: Yes

2. Has the statistical analysis been performed appropriately and rigorously? 

Reviewer #1: Yes

3. Have the authors made all data underlying the findings in their manuscript fully available?

Reviewer #1: Yes

4. Is the manuscript presented in an intelligible fashion and written in standard English?

Reviewer #1: Yes

5. Review Comments to the Author

Reviewer #1: Dear Dr,

I have gone through the manuscript and feel the manuscript was well written by the authors in this work on French validation of the Weight Efficacy Life-Style questionnaire (WEL): Links with mood, self-esteem and stress among the general population and a clinical sample of individuals with overweight and obesity. However, a minor revision is needed to improve the quality of this paper.

Abstract: In the design, the authors mentioned multiple statistical analyses. Please report the most important/significant; this will improve the result section, which in the current version is ard to understand where the major benefits happened.

Introduction:

Please clarify following question into Introduction section:

• What is the main question addressed by the research?

• How original is the topic? What does it add to the subject area compared with other published material?

• Please show the gap clearly. What is already known and what new knowledge this work will provide?

Discussion:

What are the policy implications of these findings? How can your findings inform improvements in existing nutritional interventions? What is the barrier to incorporating its use in public health?

If a public health manager wants to scale up this intervention, would this be feasible?

6. PLOS authors have the option to publish the peer review history of their article (what does this mean?). If published, this will include your full peer review and any attached files.

Reviewer #1: No

---

## [Author Response · Author response to Decision Letter 0]

29 Sep 2021

Manuscript PONE-D-21-08991

Response to Reviewers

Dear Dr. Palazón-Bru,

Thank you for giving us the opportunity to submit a revised draft of the manuscript “French validation of the Weight Efficacy Life-Style questionnaire (WEL): Links with mood, self-esteem and stress among the general population and a clinical sample of individuals with overweight and obesity” for publication in the PLOS ONE journal. 

We appreciate the time and effort that you and the reviewer dedicated to providing feedback on our manuscript and are grateful for the insightful comments on and valuable improvements to our paper.

We have incorporated all the suggestions made by the reviewer. Those changes are highlighted in red within the document named “Revised Manuscript with track Changes”. 

Please see below, in blue, for a point-by-point response to the reviewers’ comments and concerns. All line numbers refer to the revised manuscript file with tracked changes.

Sincerely, 

Natalija Plasonja

PhD student at the University of Bordeaux, France

natalija.plasonja@u-bordeaux.fr

 

Journal Requirements:

Response: The style requirements of the manuscript have been verified.

Response: Thank you for pointing this out. All the references were converted to the “Vancouver” style.

4. Please provide additional details regarding participant consent. In the ethics statement in the Methods and online submission information, please ensure that you have specified (1) whether consent was informed and (2) what type you obtained (for instance, written or verbal, and if verbal, how it was documented and witnessed). If your study included minors, state whether you obtained consent from parents or guardians. If the need for consent was waived by the ethics committee, please include this information.

Response: Thank you for the reminder. Information regarding the informed consent of patients and their legal representative are included in the Methods section, 2.1.1. ESTEAM cohort (lines 173-180). 

Response: Thank you for pointing this out. Information regarding the anonymization of the data have been integrated in the Methods section, 2.1.1. ESTEAM cohort (lines 173-180).

The authors would like to give special thanks to Michel Dubourdeaux (R&D Innovation Director at PiLeJe), Christophe Guyomar (Chief Digital Officer at PiLeJe) for their administrative and technical support and Claude Blondeau (Head of Scientific Communication at PiLeJe) for editing the manuscript. Thanks are also addressed to Dr Dominique Périn Calvão for the translation of the MDI, Dr Marilou Bruchon-Schweitzer, Dr Eve Villemur and Dr Gérard Ostermann for their contribution to the translation of the WEL and SSES scales. Anna Brytek-Matera acknowledges financial support by the „Excellence Initiative – Research University” program for years 2020-2026 of the University of Wroclaw.

Anna Brytek-Matera acknowledges financial support by the „Excellence Initiative – Research University” program for years 2020-2026 of the University of Wroclaw.

Response: Thank you for pointing this out. The funding statement was removed from the Acknowledgments section and the manuscript and was added to the updated version of the cover letter, which was then completed with additional information regarding the funding of this research. 

Response: The full ethics statement was added in the Methods section, 2.1.1. ESTEAM cohort (lines 173-180). 

 

Review Comments to the Author

Reviewer #1: Dear Dr,

I have gone through the manuscript and feel the manuscript was well written by the authors in this work on French validation of the Weight Efficacy Life-Style questionnaire (WEL): Links with mood, self-esteem and stress among the general population and a clinical sample of individuals with overweight and obesity. However, a minor revision is needed to improve the quality of this paper.

Response: Thank you very much. 

Abstract: In the design, the authors mentioned multiple statistical analyses. Please report the most important/significant; this will improve the result section, which in the current version is hard to understand where the major benefits happened.

Response: Thank you for your comment. Your suggestion was taken into account and the abstract was revised accordingly (lines 31-36). 

Introduction:

Please clarify following question into Introduction section:

• What is the main question addressed by the research?

Response: Thank you for pointing this out. The main question addressed by the research has been made clearer at the end of the Introduction section (lines 142-147).

• How original is the topic? What does it add to the subject area compared with other published material?

Response: This suggestion has been taken into account and additional information has been provided (lines 142-147). 

• Please show the gap clearly. What is already known and what new knowledge this work will provide?

Response: Thank you for your suggestion. Additional information has been included in the Introduction section (lines 142-147).

Discussion: 

What are the policy implications of these findings? 

Response: Our findings may feed into the formulation of the national health policy and strategies in obesity prevention and intervention efforts. This idea has briefly been addressed at the beginning of the last paragraph of the Discussion (lines 674 – 677).

How can your findings inform improvements in existing nutritional interventions? 

What is the barrier to incorporating its use in public health?

Response: These two questions have been addressed in the last paragraph of the Discussion (lines 674 – 683).

If a public health manager wants to scale up this intervention, would this be feasible?

Response: The public health manager needs a multidisciplinary team for creating and implementing the nutritional intervention.

---

## [Decision Letter · Decision Letter 1]

29 Oct 2021

French validation of the Weight Efficacy Life-Style questionnaire (WEL): Links with mood, self-esteem and stress among the general population and a clinical sample of individuals with overweight and obesity

PONE-D-21-08991R1

Dear Dr. Plasonja,

We’re pleased to inform you that your manuscript has been judged scientifically suitable for publication and will be formally accepted for publication once it meets all outstanding technical requirements.

Kind regards,

Antonio Palazón-Bru, PhD

Academic Editor

PLOS ONE

Additional Editor Comments (optional):

Reviewers' comments:

Reviewer's Responses to Questions

**Comments to the Author**

1. If the authors have adequately addressed your comments raised in a previous round of review and you feel that this manuscript is now acceptable for publication, you may indicate that here to bypass the “Comments to the Author” section, enter your conflict of interest statement in the “Confidential to Editor” section, and submit your "Accept" recommendation.

Reviewer #1: All comments have been addressed

2. Is the manuscript technically sound, and do the data support the conclusions?

Reviewer #1: Yes

3. Has the statistical analysis been performed appropriately and rigorously? 

Reviewer #1: Yes

4. Have the authors made all data underlying the findings in their manuscript fully available?

Reviewer #1: Yes

5. Is the manuscript presented in an intelligible fashion and written in standard English?

Reviewer #1: Yes

6. Review Comments to the Author

Reviewer #1: This is an interesting study and the authors have collected a unique dataset using appropriate methodology. The paper is generally well written and structured. The authors replied to all comments well. So, this manuscript can be accepted for publication in this journal.

7. PLOS authors have the option to publish the peer review history of their article (what does this mean?). If published, this will include your full peer review and any attached files.

Reviewer #1: No

---

## [Editor Report · Acceptance letter]

5 Nov 2021

PONE-D-21-08991R1 

French validation of the Weight Efficacy Life-Style questionnaire (WEL): Links with mood, self-esteem and stress among the general population and a clinical sample of individuals with overweight and obesity 

Dear Dr. Plasonja:

I'm pleased to inform you that your manuscript has been deemed suitable for publication in PLOS ONE. Congratulations! Your manuscript is now with our production department. 

Kind regards, 

on behalf of

Dr. Antonio Palazón-Bru 

Academic Editor

PLOS ONE